# Compositional Variations between Adult and Infant Skin Microbiome: An Update

**DOI:** 10.3390/microorganisms11061484

**Published:** 2023-06-02

**Authors:** Barry Murphy, Michael Hoptroff, David Arnold, Andrew Cawley, Emily Smith, Suzanne E. Adams, Alex Mitchell, Malcolm J. Horsburgh, Joanne Hunt, Bivash Dasgupta, Naresh Ghatlia, Samantha Samaras, Ashely MacGuire-Flanagan, Kirti Sharma

**Affiliations:** 1Unilever Research & Development, Port Sunlight, Bebington, Wirral CH63 3JW, UK; 2Eagle Genomics, Wellcome Genome Campus, Hinxton, Cambridge CB10 1DR, UK; 3Institute of Infection Biology, Veterinary and Ecological Sciences, University of Liverpool, Liverpool L69 7ZB, UK; 4Unilever, 40 Merritt Blvd, Trumbull, CT 06611, USA; 5Unilever, North Rocks Road, North Rocks, NSW 2151, Australia

**Keywords:** microbiome, skin, infant, adult, networks, function

## Abstract

Human skin and its commensal microbiome form the first layer of protection to the outside world. A dynamic microbial ecosystem of bacteria, fungi and viruses, with the potential to respond to external insult, the skin microbiome has been shown to evolve over the life course with an alteration in taxonomic composition responding to altered microenvironmental conditions on human skin. This work sought to investigate the taxonomic, diversity and functional differences between infant and adult leg skin microbiomes. A 16S rRNA gene-based metataxonomic analysis revealed significant differences between the infant and adult skin groups, highlighting differential microbiome profiles at both the genus and species level. Diversity analysis reveals differences in the overall community structure and associated differential predicted functional profiles between the infant and adult skin microbiome suggest differing metabolic processes are present between the groups. These data add to the available information on the dynamic nature of skin microbiome during the life course and highlight the predicted differential microbial metabolic process that exists on infant and adult skin, which may have an impact on the future design and use of cosmetic products that are produced to work in consort with the skin microbiome.

## 1. Introduction

The human skin is the largest and outermost organ with diverse and multiple roles. Subject to daily challenge, the skin encounters and defends against environmental insults from friction, abrasion and UV light, as well as the application of personal care products. The human skin structural maturation progresses across the life course. Newborns have an epidermal thickness that is approximately 30% less than adult skin [1]. There are multiple differences in the physiochemical properties of juvenile skin compared with adults. Infant skin shows increased levels of trans-epidermal water loss (TEWL) [2,3], and elevated levels of sebum that subsequently decrease to a level lower than that of adult skin after 6 months [4,5]. At birth, the skin surface pH is neutral or alkaline (pH 6.2–7.5) [6,7] then decreases in the first 4 weeks of life to a pH of 5–5.5, similar to that of older children and adults [8,9,10]. The acidity of the skin surface influences the cutaneous bacterial composition [11,12,13] that acts as an important defense mechanism against pathogens [9,14].

The microbial inhabitants of skin, collectively termed its microbiota, provide an essential part of skin’s key functions. Comprising bacteria, archaea, fungi, mites and viruses [15,16,17], these biota contribute to the structure and function of skin in both healthy and diseased states. Microbiomes described as being in dysbiosis have been identified in skin conditions ranging from atopic dermatitis [18,19] to dandruff [20,21] and axillary malodor [22,23], with causative relationships established in many instances.

The skin microbiome composition, and its underlying functionality, is a consequence of the local ecology of a particular body site with distinct microbiomes present at moist (e.g., axilla), sebaceous (e.g., face) and dry/sebaceous gland poor (e.g., forearm or leg) sites [24,25]. In reality, the stark variation in the underlying nutrient and lipid availability, pH and water activity (a_w_) and the resultant commensal microbiome, means the skin microbiome could be considered to be a collection of discrete microbiomes, each with their own unique characteristics [15,25].

The skin microbiome of infants, like skin structure, has been shown to be dynamic and can be impacted by multiple factors as it matures over time. Multiple external factors can contribute to the establishment and development of the skin microbiome. The initial skin microbiome composition may be impacted by the mode of delivery [26] and the mode of feeding [27]; however, multiple studies support that this initial variation in bacterial abundance between natural and cesarian-section-delivered children becomes normalized in the first few months of life, with body-site-specific microbiomes becoming evident, as seen in adult subjects [28,29,30]. Additional factors beyond the mode of birth that can impact the infant skin microbiome composition have previously been reported, including household composition [31] and the use of skin care products [32]. 

The infant skin microbiome, after an initial variability potentially impacted by, e.g., the method of delivery, becomes a surface dominated by bacteria of the genus *Streptococcus*, and to a lesser extent, the Gram-positive genus *Gemella*, and the Gram-negative genera *Prevotella* and *Haemophilus*, creating a skin microbiome characterized by elevated (relative to adult) levels of taxa more commonly associated with the oral microbiome [33]. The abundance of these genera decreases over time as the microbiome transitions to one more associated with adult skin [29,34]. The lower relative abundance of the adult skin commensal *Cutibacterium acnes* was explained as a result of limited maturation of sebaceous glands, the primary source of nutrients for lipophilic bacteria, including *Cutibacterium*, *Corynebacterium* and the fungi *Malassezia* [16,35,36]. This correlation of sebaceous gland activity and microbiome alteration is additionally seen in post-menopausal individuals where microbiome diversity changes and composition reversion are in evidence following a reduction in sebaceous gland excretion [37,38,39].

Here, we present a comparison of healthy infant and adult skin microbiomes from unrelated subjects examining variations in taxonomic composition, diversity, network connectivity and the predicted functional capacity between the life stages. 

## 2. Materials and Methods

### 2.1. Ethics Statement

Written informed consent was obtained from all the enrolled individuals or parents/guardians. The study protocol was approved by the IntegReview Independent Ethics Committee (Infant Cohort) or the Institutional Review Board Services (Adult Cohort). The methods were carried out in accordance with the principles of the Declaration of Helsinki and Good Clinical Practice as applicable to clinical studies on cosmetics.

### 2.2. Study Participants

The study samples were pooled for meta-analysis from 2 independent internal studies. All samples were taken from subjects in the USA. A total of 120 samples, split between groups evenly, were selected from an infant cohort (Montana, USA, mean age 11.6 months, range 6–18 months) and an adult cohort (Texas, USA, mean age 31.85 years, range 21–50 years). None of the adult subjects were related to the subjects in the infant cohort, so an examination of familial links was not possible. All subjects were recruited based on having healthy skin at the proposed sampling sites, having no underlying health conditions and not using any medication known to impact the skin microbiome for the previous 3 months. Samples, taken from the legs of subjects in both studies were collected, stored, processed, sequenced and analyzed in an identical fashion, minimizing any potential bias as a result of sample collection methodologies or data processing variations.

### 2.3. 16S rRNA Gene Sequencing

#### 2.3.1. Microbiome Sample Collection and Processing

Buffer washes were collected from all participants (leg skin) using a sterile Teflon sampling ring (internal diameter 3.5 cm) using the cup scrub method [40] as previously described [20]. Next, 2.0 mL of buffer wash solution (sterile phosphate-buffered saline pH 7.9 containing 0.1% *v*/*v* Triton X-100) was pipetted into the sampling ring and the skin surface gently agitated for 1 minute with a sterile Teflon rod. The sampling fluid was collected using a new sterile disposable pipette and placed into a sterile centrifuge tube. The sampling procedure was repeated with a further 2.0 mL aliquot of buffer wash material and both aliquots pooled. The samples were placed on ice during the collection process and then stored at −80 °C prior to the DNA extraction. The shipment of samples from both studies, prior to extraction, was carried out on dry ice with appropriate temperature logging.

#### 2.3.2. DNA Extraction and 16S rRNA Gene Library Preparation and Sequencing

DNA extractions were carried out as previously described [41]. Library prep was carried out using oligonucleotide primers targeting the V1-V2 hypervariable region of the 16S rRNA gene,

U28F: 5′ACACTCTTTCCCTACACGACGCTCTTCCGATCTNNNNNAGAGTTTGATCMTGGCTCAG3′

U338R: 5′GTGACTGGAGTTCAGACGTGTGCTCTTCCGATCTTGCTGCCTCCCGTAGGAGT3′

The primer pair were modified versions of the standard 28F and 338R primers containing additional recognition sequences to facilitate nested PCR using methods described previously [42]. 

Second-round PCR incorporated Illumina adapters containing indexes (i5 and i7) for sample identification utilizing eight forward primers and twelve reverse primers, each of which contained a separate barcode allowing up to 96 different combinations. The general sequences of the primers are illustrated below with the variable 8 bp barcode underlined; the amplification was performed as previously described [42].

N501f: 5′AATGATACGGCGACCACCGAGATCTACAC*TAGATCGC*ACACTCTTTCCCTACACGACGCT3′

N701r: 5′CAAGCAGAAGACGGCATACGAGAT*TCGCCTTA*GTGACTGGAGTTCAGACGTGTGCTC3′

#### 2.3.3. Informatics Processing

The sequence analysis was performed using the QIIME2 microbiome analysis tool suite [43] version 2019.1. The paired-end sequences were imported into a QIIME2 format, then denoised using DADA2 [44]. The primer sequence regions were removed during denoising by setting the DADA2 forward and reverse read trim parameters to the length of the forward and reverse primers, respectively. A complete list of the software parameters and versions can be found in Appendix A. Denoising produced 40,054 unique amplicon sequence variants (ASVs) corresponding to 32,091,435 sequences in the 120 original samples. The samples were rarefied to 10,000 reads per sample in advance of the diversity analysis. Rooted and unrooted phylogenetic trees were generated for the ASVs using the QIIME2 phylogeny align-to-tree-mafft-fasttree workflow. The taxonomy assignments were generated by comparing the ASVs against a BLAST database composed of the Human Oral Microbiome Database (HOMD), HOMD extended and GreenGenes sequences (HOMDEXTGG version 14.51) described in [45]. The taxonomic classification was performed as previously described [46] at a 99% identity across 98% of the read length. Metagenomic functional predictions were carried out using PICRUSt2 v2.4.2 [47,48,49,50,51]. QIIME2 derived ASV tables in biom format were utilized. Default settings using the PICRUSt2 reference database and assignment against the MetaCyc pathway database were carried out [52].

### 2.4. Data Processing

#### 2.4.1. Network Analysis

A co-occurrence network analysis was carried out on QIIME2 generated ASV tables collapsed at the species level. In such methods, the sparsity of microbial datasets often results in spurious correlations. To address this, the taxa were filtered based on 0.05% relative abundance and ASVs were removed when present in fewer than 37% of the samples, in line with previous studies. The networks were inferred using Sparse InversE Covariance estimation for Ecological Association and Statistical Inference (SPIEC-EASI version 1.0.7) and at the prevalence threshold selected, the networks appeared stable across a range either side [53]. The neighborhood method was chosen and the StARS (Stability Approach to Regularization Selection) method used with a lambda max threshold of 0.01. The visualizations of the networks were produced in Cytoscape (version 3.7.2) [54].

#### 2.4.2. Statistical Analysis

Statistical analysis of 16S rRNA gene metataxonomic data including alpha and beta diversity analysis was carried out using the available scripts in QIIME2 version 2019.1. Within sample group diversity (alpha) changes (Observed and Shannon) were estimated and tested using non-parametric approaches. A signed rank test for changes across time-points for each treatment that accounts for paired differences within subjects. Kruskal–Wallis tests were used for pairwise treatment comparisons. Between group diversity (beta) was assessed visually using Non-Metric Multidimensional Scaling (NMDS) ordination plots for key metric distance matrices, Bray–Curtis (semi-metric), Jaccard, weighted and unweighted Unifrac [55]. The statistical inference was achieved using permutation analysis of variance (PERMANOVA). Differentially abundant taxa were identified using Linear discriminant analysis Effect Size (LEfSe) [56] and explainable AI machine learning methods, which were visualized via SHAP (SHapley Additive exPlanations) analysis [57]. 

## 3. Results

### 3.1. Bacterial Diversity Analysis

The raw DNA sequences were processed using QIIME2 as per the Methods section, producing ASV level tables that were used for alpha diversity analysis. The analysis was carried out on both groups following rarefaction of all the samples to a minimum depth of 10,000 reads per sample. Alpha diversity analysis, using both Observed and Shannon metrics, showed contrasting results. Observed alpha diversity (Richness), Figure 1a, was significantly elevated in adult skin microbiome samples (*p* < 0.05), whereas Shannon diversity (Richness and evenness), Figure 1b, was significantly elevated in infant microbiome samples (*p* < 0.05).

### 3.2. Taxonomic Composition and Differential Abundance

ASVs were classified against a BLAST database composed of the Human Oral Microbiome Database (HOMD), HOMD extended and GreenGenes sequences as described in the Methods section. This analysis produced relative abundances of species level taxa for all samples. The taxonomic abundances of the major community members of both the infant and adult skin microbiomes were assessed to ascertain community profiles (Figure 2a,c). Significant differences in mean relative abundances, as calculated using LEfSe (LDA > 2.5 and *p* < 0.05), were identified (Figure 2b (genus) and Figure 2d (species)). Relatively, several genera were elevated in their abundances on infant skin including *Streptococcus*, *Acinetobacter* and *Neisseria* whereas *Cutibacterium*, *Lactobacillus*, *Micrococcus*, *Enhydrobacter* and *Finegoldia* were elevated on adult skin. From the species determinations, multiple members of the genus *Streptococcus* including *S. thermophilus*, *S. vestibularis*, *S. mitis*, *S. crispatus*, *S. peroris* and *S. salivarius* were elevated in their relative abundance levels on infant skin. Whereas on adult skin *Cutibacterium acnes*, *Micrococcus luteus*, *Lactobacillus crispatus* and *Lactobacillus iners* were identified as having elevated abundance in comparison to infant skin. 

The differentially abundant taxa were highly prevalent across all samples and differential abundance was not driven by a subset of samples with high abundance of target taxa or the absence of these taxa in samples (Appendix A).

In addition to the standard differential abundance analysis of the DNA sequence data, machine learning methods (random forest decision trees) [58] were employed to identify the bacterial taxa that were indicative of skin microbiomes from babies and adults. SHapley Additive exPlanations (SHAP) analysis was used to visualize differential taxa and can been seen in Figure 3a (genus) and Figure 3b (species). Each dot represents a microbiome sample and the corresponding taxon abundance in that sample for genus or species. Red dots depict a taxon that is enriched and blue dots indicate there is reduced abundance. Clusters of red dots indicate an increased abundance of that taxon corresponding to the life stage on the *y* axis. 

There was a high degree of correlation between taxa identified via machine learning in comparison to LEfSe. Relative abundances of certain genera, including *Streptococcus* and *Cutibacterium*, were highly indicative of a microbiome originating from infant or adult skin, respectively. Infant skin was additionally characterized by other genera associated with the oral microbiome including *Rothia*, *Gemella*, *Granulicatella* and *Neisseria*. Machine learning analysis indicated that on adult skin, there were elevated levels of Gram-positive Anaerobic Cocci (GPAC), including *Finegoldia*, *Anaerococcus* and *Peptoniphilus*. The results at a species level were consistent for the source microbiome tested in the study and showed multiple species of *Streptococcus* were indicative of infant skin and *Cutibacterium acnes* was a strong biomarker of adult skin.

### 3.3. Beta Diversity Analysis

Beta diversity analysis was carried out to examine the differences between the life stage groups at a community level. The analysis was carried out using both non-phylogenetic (Bray–Curtis and Jaccard) and phylogenetic (Weighted and Unweighted Unifrac). As with alpha diversity analysis, all samples were rarefied to a minimum depth of 10,000 reads per sample. Statistical inference was achieved using a permutation analysis of variance (PERMANOVA) and demonstrated significant differences (*p* < 0.05) for all metrics when comparing the adult and infant skin microbiome samples. The data were visualized using non-metric multidimensional scaling ordination plots. The distinct clusters for both infant and baby groups are visible for all the metric analysis, representing significant differences in community composition between groups on both a taxonomic abundance and phylogenetic level with centroids for each group included (Figure 4a–d). 

### 3.4. Functional Analysis

The predicted functional analysis of the skin microbiome was carried out using PICRUSt2 [47]. The differentially abundant pathways were identified using LEfSe (LDA > 2.75, *p* < 0.0001). The functional biomarkers that were differentially abundant between the groups were visualized (Figure 5) on both a group and per sample basis. The multiple pathways associated with the generation of lipids on skin were increased on infant skin whereas pathways indicative of adult skin included acid production (propionate) as well as amino acid and sugar metabolism. The lipid production pathways were predicted based on the elevated abundance of streptococci with propionic acid production, as expected, associated with increased levels of *Cutibacterium acnes*.

### 3.5. Network Analysis

The network analysis was carried out to examine co-occurrence connections in the infant and adult skin microbiomes. Network analysis has been used extensively to examine and infer inter-species interactions in a community using methods including correlation analysis, hierarchical modelling and linear regression [59]. Analysis using Sparse InversE Covariance estimation for Ecological Association and Statistical Inference showed an increased level of network connectivity was evident in the infant skin microbiome both at the community level and visualized using Cytoscape, (Figure 6a,b). 

Whereas the nature of the interactions between the relevant taxa differs between the groups, more positive network interactions between the dominant genus (*Streptococcus*) and minor community members are observed. The networks constructed from adult skin microbiome data show more balanced levels of positive and negative interactions between taxa. The network analysis has been used for the skin microbiome in previous studies to show improvements in skin condition following treatment [60] and the impact of pollution on the skin microbiome [61]. In these examples, increased network connectivity has been associated with increased health. In this instance, however, both infant and adult groups were recruited based on having healthy skin free of cosmetic issues, which can in part explain no significant differences in the network connectivity.

## 4. Discussion

Next-generation sequencing analysis was used to examine the microbiome composition of the infant and adult skin microbiomes. The subsequent taxonomic and diversity analysis of these data revealed pronounced differences in the community composition. 

Community diversity analysis highlighted significant differences between the infant and adult skin microbiomes. Alpha diversity analysis showed increased richness (Observed diversity) in adult skin, however, analysis of richness and abundance (Shannon diversity) showed elevated levels on infant skin. The reduction in alpha diversity on adult skin has previously been attributed to the increase in sebaceous gland activity and subsequent abundance of sebum as a primary food source for skin organisms. The associated increase in *C. acnes* levels leads to a decrease in Shannon alpha diversity [33]. Indeed, an increase in skin alpha diversity has been seen in post-menopausal women following a decrease in sebaceous gland activity and subsequent reduction in skin sebum levels [38]. Significant differences were also seen in beta diversity analyses for all four-core metrics used (Bray–Curtis, Jaccard, Weighted and Unweighted Unifrac). Taken together, these data demonstrate that the number, abundance and phylogenetic classification of both the dominant and minor community members are significantly different between the infant and adult skin microbiomes. 

Taxonomic differences were examined using machine learning methods to gain insights into the compositional differences between host microbiotas. The approach used here validated the Linear discriminant analysis and the Effect Size analysis to support that there were differentially abundant genera and species between adult and infant microbiota. The infant skin microbiome is characterized by elevated levels of streptococcal species including *S. thermophilus*, *S. vestibularis*, *S. infantis* and *S. mitis* along with four others. These streptococci represent the dominant lactic acid bacteria on infant skin and their increased abundance and presumed metabolic activity may contribute to the acidification of infant skin following birth. At birth, potentially as a consequence of exposure to alkaline amniotic fluid, an infant’s skin pH is close to neutral but acidifies in the months after birth, forming the acid mantle [62]. With streptococcal species accounting for approximately one quarter of the bacterial microbiome, it can be hypothesized that the abundance of these lactic acid producers could drive this acidification process. 

In contrast, the adult skin microbiome is dominated by the lipophilic bacterium *C. acnes*. While the average abundance of *C. acnes* (33%) on the leg site sampled, is lower than what would be seen on a sebaceous skin site, e.g., the face (>70%) [63], the elevated levels seen on adult skin are consistent with other analyses that ascribe the alterations to the impact of increased sebaceous secretions on the skin during and following puberty [16,34]. Beyond *C. acnes*, the adult skin microbiome has a discrete profile of species representing the lactic acid bacteria. In comparison to infant skin, dominated by streptococci, adult skin is alternatively colonized by multiple species of *Lactobacillus* including *L. crispatus* and *L. iners*. There is currently not a clear explanation for this change in lactic acid bacteria, however, potential community interactions with *C. acnes* and the associated increase in the levels of sebum derived antimicrobial lipids may be drivers for the identified taxonomic abundances. Of note, levels of *Streptococcus* are elevated on the skin of post-menopausal individuals, suggesting that sebaceous lipids and the subsequent breakdown products may indeed be important in controlling the abundance of streptococci on the skin [57,64]. 

Machine learning analysis of the microbiome data additionally highlighted elevated levels of Gram-positive Anaerobic Cocci (GPAC) bacteria on adult skin. A previous connection has been made between GPAC and the levels of filaggrin expression in skin, with GPAC abundance reduced in filaggrin deficient skin [65]. Reduced filaggrin expression is a significant risk factor for atopic dermatitis development [66]; however, in our study all subjects, babies and adults, were classified as healthy individuals, free of any cosmetic skin conditions. Previous analysis identified reduced levels of filaggrin derived natural moisturizing factors (histidine, pyrrolidone carboxylic acid and urocanic acid) at various body sites of babies, which at least in part may explain the reduced levels of GPAC observed on infant skin [67].

The predicted functional differences between the infant and adult skin microbiome were identified using PICRUSt2. The functional potential of the infant skin microbiome generally shows increased levels of lipid biosynthesis associated with the increased levels of streptococci on the skin. Differentially elevated pathways for the biosynthesis of gondoate, palmitate and oleate as well as pathways for (5Z)-dodec-5-enoate biosynthesis indicate an increased level of cosmetically relevant microbial fatty acid biosynthesis on the skin. Indeed, the additional pathways for fatty acid elongation and biosynthesis of L-isoleucine, a branched chain amino acid that is a precursor for ceramide biosynthesis, are also elevated, offering the intriguing possibility of microbially derived fatty acid synthesis providing skin benefits. On adult skin, the production of propanoate was predicted to be elevated in accordance with the increased levels of propionic acid producing *Cutibacterium acnes*. Additionally, an elevated level of heme biosynthetic pathways is seen on adult skin with these pathways more likely directed to the production of protoporphyrin IX, uroporphyrin III and coproporphyrin III, all porphyrins produced by *C. acnes* [68]. The superpathway of heme biosynthesis from glutamate is likely associated with the elevated L-histidine degradation I pathways, one endpoint of which is the production of glutamate. In contrast to infant skin, where lipid biosynthetic pathways were elevated, differentially elevated pathways on adult skin are associated with sugar degradation for bacterial energy production via the TCA cycle including beta D-glucuronide and D-glucuronate degradation, inositol degradation and D-fructuronate degradation. Recent matched microbiome and metabolome analysis has been carried out on an infant skin cohort and has identified distinct cluster correlations between commensal microbiota and metabolite classes including *Cutibacterium* with hydrophobic skin barrier components and *Staphylococcus* with amino acids [69]. This work also identifies the correlation of *Cutibacterium* with the pathways associated with sugar and lipid metabolism similar to the data presented here; however, the direct measurement of metabolites by Roux et al. provides more direct evidence between the microbial metabolism and skin metabolome content.

Co-occurrence network analysis was carried out to examine the relationships between the bacterial community members in each group. The network analysis was carried out using the species assigned to the top 10 most abundant genera in each group, with the infant skin microbiome having fewer network connections overall than the adult skin microbiome, though not statistically significant based on bootstrapping analysis. This could be explained due to both study groups having underlying healthy skin in comparison to previous uses of these approaches comparing health and disease. Within this sparser network, however, 30% of the inter-node interactions are mediated by *Streptococcus* species, the majority of which are positive interactions with other community members. In comparison, the adult skin network shows a greater balance of positive and negative interactions with a reduced importance on *Streptococcus* mediated interactions and an increased importance on *Staphylococcus* and *Lactobacillus* interactions.

The increased number of *Streptococcus* interactions could suggest a heightened importance of *Streptococcus* metabolite exchange between the community members, highlighting a keystone genus role in community functionality. *Streptococcus* species were shown to have a variety of beneficial characteristics associated with skin health. As well as being lactic acid, peroxide [70] and bacteriocin [71] producers, preventing the growth of potential pathogens including *S. aureus*, streptococci increase ceramide production in skin following topical application [72,73] as well as the production of hyaluronic acid [74], both key ingredients in skin care compositions. The potential importance of *Streptococcus* spp. as key members of the infant skin microbiome and potentially the transition to an adult community highlights the need to be cognizant of the design of cosmetic products specifically formulated for infant skin, as opposed to adult skin, and their potential impact on the skin microbiome. While this investigation provides information on the infant and adult skin microbiomes, there are some limitations to the study. 

The focus of this work has solely been the bacterial composition of the microbiome. Shotgun metagenomics analysis of the skin microbiome would provide information not only on the bacterial composition, but also on the fungal and potential viral composition of the microbiome. It would be expected that variations would be present in the fungal mycobiome due to variations in sebaceous lipids on skin between infants and adults, which would impact the levels of Malassezia spp. on skin as described previously [75,76]. Additionally, due to the lack of relationship between infants and adults in this study, it was not possible to examine the similarities or differences between related and non-related individuals as previous studies [29]. Further investigations should encompass related and unrelated subjects as well as more comprehensive microbiome profiling, including the quantitative assessment of target genera and species via qPCR and additional microbiome related measures such as lipid and metabolite analysis to investigate if the predicted differences in functional pathway analysis are borne out in direct measurement data. 

## 5. Conclusions

Significant differences in community composition, connectivity and functional potential exist between the infant and adult skin microbiome. Differences also exist between the underlying stratum corneum, however, there is limited understanding of the evolution of the infant skin microbiome and whether this evolution is partly driven by microbiome composition. A more in-depth understanding of skin microbiome dynamics across the life course is warranted in order to understand if the early life skin microbiome can impact skin health later in life. 

## Figures and Tables

**Figure 1 microorganisms-11-01484-f001:**
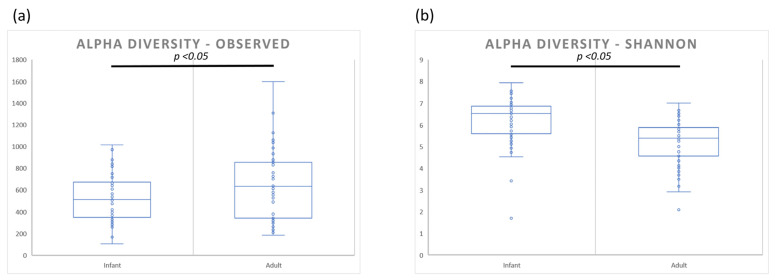
Diversity analysis of the adult and infant skin microbiome: alpha diversity analysis of the adult and infant skin microbiome using (**a**) Observed and (**b**) Shannon alpha diversity metrics. Data are visualized using box and whisker plots with significant differences for alpha diversity calculated using Kruskal–Wallis pairwise comparisons with significant threshold set at *p* < 0.05. Significant differences were observed between adult and infant groups for both metrics.

**Figure 2 microorganisms-11-01484-f002:**
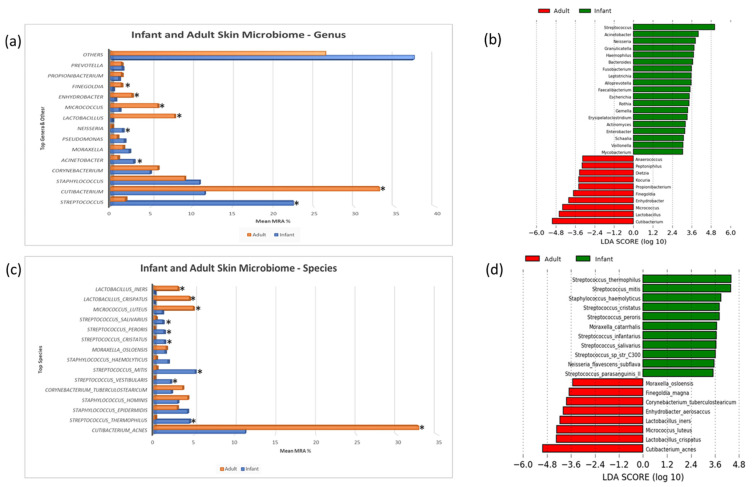
Taxonomic assessment of the infant and adult skin microbiomes: a and c, bar plot representations of the most abundant genera (**a**) and species (**c**) identified in the adult and infant skin microbiome with differentially abundant genera and species highlighted by * (*p* <0.05). (**b**,**d**) Differentially abundant genera (**b**) and species (**d**) identified between the infant and adult skin microbiome as identified using Linear discriminant analysis Effect Size (LEfSe) with a statistical significance cut off set at LDA > 2.5 and *p* < 0.05.

**Figure 3 microorganisms-11-01484-f003:**
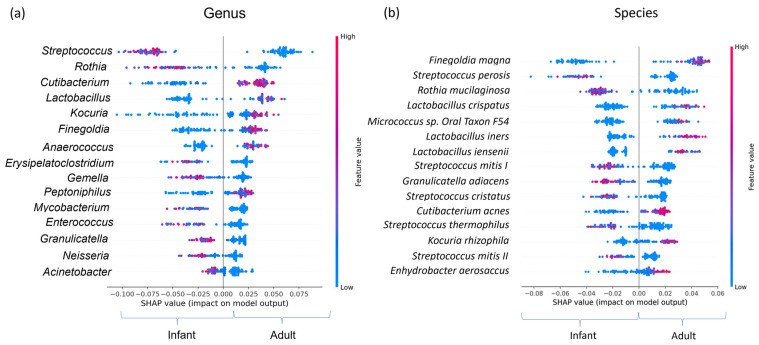
Machine learning analysis of the infant and adult skin microbiome: SHapley Additive exPlanations (SHAP) summary dot plots as computed using an optimized machine learning model on features (ASVs) data. Each dot represents a microbiome sample and the corresponding taxon relative abundance in that sample for genus (**a**) or species (**b**). The red dots depict a taxon that is enriched, and the blue dots indicate there is reduced abundance. The clusters of red dots indicate an increased abundance of that taxon corresponding to the life stage on the *y* axis.

**Figure 4 microorganisms-11-01484-f004:**
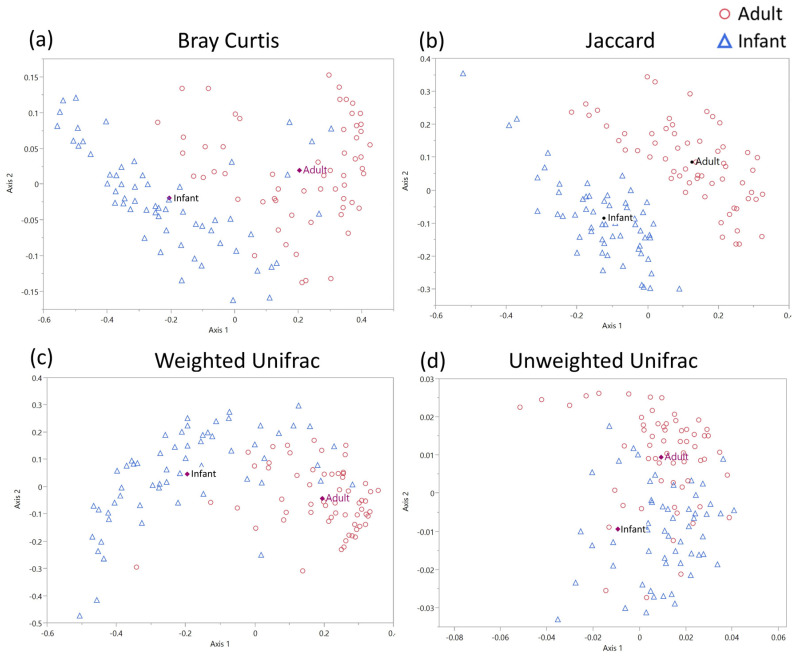
Beta diversity analysis of the adult and infant skin microbiome: beta diversity analysis of the adult and infant skin microbiome using (**a**) Bray–Curtis, (**b**) Jaccard, (**c**) Weighted Unifrac, (**d**) Unweighted Unifrac beta diversity metrics. Data are visualized using non-metric multidimensional scaling ordination plots with significant differences in beta diversity calculated using pairwise PERMANOVA comparisons with significant threshold set at *p* < 0.05.

**Figure 5 microorganisms-11-01484-f005:**
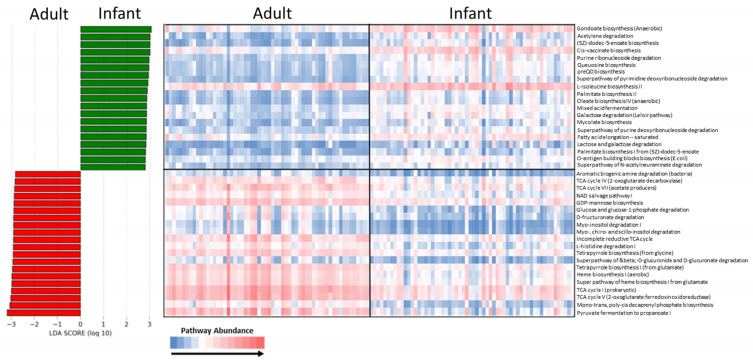
Differentially abundant predicted functional pathways between the adult and infant skin microbiomes: functional biomarkers elevated in the adult and infant skin microbiomes, calculated using LEfSe, are highlighted with red and green bars, respectively. LDA > 2.75, *p* < 0.0001. The heatmap visualizes the per sample pathway abundance identified as being differentially abundant between the sample groups with LefSe.

**Figure 6 microorganisms-11-01484-f006:**
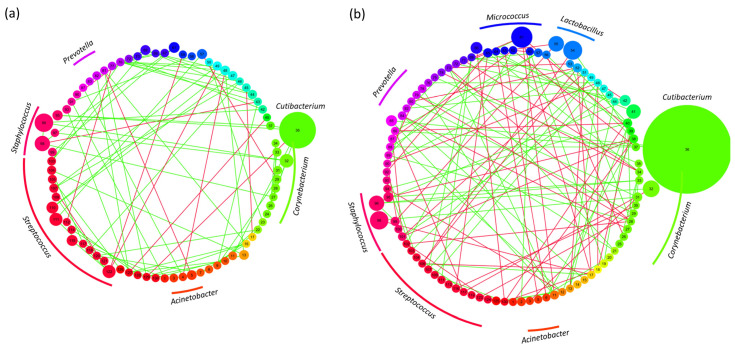
Network analysis of infant and adult skin microbiomes: cytoscape visualization of co-occurrence network connections in (**a**) infant and (**b**) adult skin microbiomes. Nodes represent individual species of bacteria (complete list of taxa can be found in Appendix A) grouped by color at the genus level with node size showing their relative abundance. All taxa below 1% abundance are represented by equal sized circles. Green lines indicate positive correlations and red lines indicate negative correlations.

## Data Availability

All sequencing data and appropriate metadata have been deposited in the SRA (Accession number PRJNA902295).

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
