# Peer review of "Compositional Variations between Adult and Infant Skin Microbiome: An Update"

_microorganisms, 2023, doi:10.3390/microorganisms11061484_

Round 1

Reviewer 1 Report

After reading the article carefully I have the following comments :

How was the sample calculated?

Why was the leg site selected to obtain the samples for study?

Pediatric samples were obtained in Montana, whereas adult samples were obtained in Texas. Since microbiome can be modified or influenced by climate, among a myriad of other factors. How can you be sure that the differences observed between children and adults do not depend on other confounders and not only in the age.

How do you explain the differences between observed alpha diversity and Shannon diversity seen in your sample?

Author Response

I would like to take the opportunity to thank this reviewer for their time in reviewing our manuscript. The prompt nature of the replies and comments always make the submission process more rewarding and myself and the other authors are pleased that the article was well received. Please find below responses to each of the individual comments made by the reviewer:

How was the sample calculated? - I believe that the reviewer is referring to the "sample size" and the calculation of such. In this meta analysis we selected all samples available to us from  2 independent studies to include in this analysis. A total of 60 samples per group were used for comparisons. We believe that this number of samples represents a sufficient number for appropriate comparisons to be carried out. These number compare well with studies such as doi:10.1016/j.jid.2019.05.018 and doi.org/10.1016/j.jid.2021.11.029., and significantly more than more recent comparisons e.g., doi:10.1016/j.jid.2021.07.159 (17 samples in total), or doi.org/10.1371/journal.pone.0192443 (23 samples in total). We believe that based on previously published analysis we have a sufficient sample size to examine the questions posed.

Paediatric samples were obtained in Montana, whereas adult samples were obtained in Texas. Since microbiome can be modified or influenced by climate, among a myriad of other factors. How can you be sure that the differences observed between children and adults do not depend on other confounders and not only in the age - This is an excellent point and one that was considered during the preparation of the manuscript. The samples used for this analysis were selected from two independent studies and combined for this meta analysis. Due to our internal analysis procedures we were able to control for sample collection, sample processing (including storage, DNA extraction and library processing) as well as data generation and analysis of the subsequent sequencing data. This in effect controlled for a major contributor to sample processing bias. When processing the analysis we were mindful of comparisons to other published data which examined the differences between adult and infant microbiomes. The degree of taxonomic concordance between our data and other published data (though we present both species and functional data as well as co-occurrence network analysis) suggested that the results were indeed based on the age differences between adults and infants. In addition as a sound biological rationale for the differences, in the main elevated production of sebum on adult skin, can in a large way account for these differences e.g., the increase in the levels of Cutibacterium, we are confident that the differences seen are age related. 

How do you explain the differences between observed alpha diversity and Shannon diversity seen in your sample? - This was an interesting observation in the data and well noted by the reviewer.  What should be noted is that where as both observed and Shannon metrics are both measure of alpha diversity both measure slightly different elements of alpha diversity. Observed diversity is a measure of species richness i.e. the number of species in a sample. As such Observed diversity is weighted based on the number of species present and as the taxonomic analysis suggests, (increase number of taxa classified "others") in the infant dataset, there is increase richness measures in the infant data. Conversely, Shannon diversity is increased in the adult samples. Shannon diversity is a measure of both richness and evenness,  and as such (unlike observed diversity) takes into account the relative abundance of each species. The dominance of C. acnes in adult skin results in a reduced Shannon diversity in adult samples thus explaining the variations between groups. This has been clarified in the manuscript.

Many thanks again to both reviewers for their comments and I am at their disposal for future revisions. 

Sincerely,

Dr Barry Murphy.

Reviewer 2 Report

Thank for this nice compairism of the infant with the adult skin. The main question addressed by the research is the compairism of the infant and adult skin microbiome (V1-V2 region 16s rRNA gene sequencing). The topic is relevant in the field. The authors should mention the body site, from which the samples were taken. Further controls should include related subjects, as mentioned in the manuscript. The conclusions are consistent with the evidence and arguments presented and address the main question posed. The figures also represented the data nicely.

- Line 66: please further explain "initial normalization".

- please highlight from which body sites the samples were taken.

Thank you.

Author Response

I would like to take the opportunity to thank this reviewer for their time in reviewing our manuscript. The prompt nature of the replies and comments always make the submission process more rewarding and myself and the other authors are pleased that the article was well received. Please find below responses to each of the individual comments made by the reviewer:

The authors should mention the body site, from which the samples were taken - The methods in the paper state that the body site the samples were taken from was the leg, for both adults and infants. However we have included additional references to what body site the samples were taken from for ease of reading.

Further controls should include related subjects, as mentioned in the manuscript - Indeed, as mentioned in the paper a limitation of this study is that the subjects (adults and infants) were not related. Having a follow up analysis of the effect of hereditary relationship as well as co-habitation would be an interesting build on this analysis. This type of "Real Family / Fake Family" approach is explored in; doi:10.1016/j.jid.2019.05.018. This work suggested that the skin microbiome of mothers is significantly more similar to that of their own children than that of unrelated children. an additional interesting follow up would be to examine the same relationship with fathers as well as mothers. In this work however, this was not possible but is a fruitful aera for further exploration. 

Line 66: please further explain "initial normalization" - This was an attempt to describe the reduction in microbiome flux that might be seen soon after birth, for example variability introduced by method/mode of delivery or location of delivery. The text has been updated to make this clearer.

Many thanks again to both reviewers for their comments and I am at their disposal for future revisions. 

Sincerely,

Dr Barry Murphy.